# Temperature Estimation of a Deep Geothermal Reservoir Based on Multiple Methods: A Case Study in Southeastern China

Wenjing Lin [1,2,]* and Xiaoxiao Yin [3]

1   Institute of Hydrogeology and Environmental Geology, Chinese Academy of Geological Sciences, Shijiazhuang 050061, China
2   Technical Innovation Center for Geothermal and HDR Exploration and Development, Ministry of Natural Resources, Shijiazhuang 050061, China
3   Tianjin Geothermal Exploration and Development Designing Institute, Tianjin 300250, China
*   Correspondence: linwenjing@mail.cgs.gov.cn

**Abstract:** Estimating deep geothermal reservoir temperatures is an essential mission of geothermal exploration and development. The thermal reservoir temperature estimated directly using geothermometry without comparative analysis is often far from the actual temperature. In this paper, taking the typical geothermal systems in the Xiamen Island–Zhangzhou area of southeastern China as an example, different methods such as a water–rock equilibrium analysis, $SiO_2$ geothermometer, multi-mineral equilibrium diagram, and silica-enthalpy mixing model are used to make a quantitative and qualitative analysis of the chemical equilibrium of minerals and fluids in the geothermal system. Finally, the applicability of different methods was compared and analyzed, and the geothermal reservoir temperature was estimated using the appropriate method. The results show that the calculated results of the Si-enthalpy mixing model of a typical geothermal system in southeastern China are significantly high. At the same time, the $SiO_2$ geothermometer (without vapor loss), which is closest to the results of the multi-mineral equilibrium diagram, was chosen as the geothermal reservoir temperature of the geothermal system in the study area. This study can provide a reference for the future selection of methods of deep geothermal reservoir temperature estimation in similar areas.

**Keywords:** geothermal system; geothermal reservoir temperature; water–rock equilibrium; mixing model; geothermometry

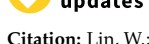



## 1. Introduction

Geothermal reservoir temperature is an important parameter that is indispensable for analyzing the type of geothermal system genesis and conducting geothermal resource evaluation. Therefore, estimating deep geothermal reservoir temperature is crucial in thermal reservoir engineering studies [1]. Two methods for determining geothermal reservoir temperature are direct measurement and estimation. Direct measurement involves drilling a well to expose or penetrate the geothermal reservoir, directly measuring the temperature in the well, and using the average of the top and bottom temperatures as the temperature of the geothermal reservoir. This method is accurate and reliable but time-consuming and expensive, and it is less easy to implement in practice, so, in most cases, the temperature of a geothermal reservoir is mainly estimated. Among the methods, the most widely used is the geothermometer, a cost-effective method [2].

After the minerals and fluids (or different fluids) in deep geothermal reservoirs reach chemical equilibrium, the geothermal reservoir temperature can be estimated from the equilibrium temperature of the chemical reaction if the chemical content remains constant during the rise of the hot water to the surface, despite the decrease in temperature. Currently, the commonly used geothermal thermometers for determining reservoir temperature include the cationic geothermometer, silica geothermometer, gas chemical geothermometer, solute isotope geothermometer, gas isotope geothermometer, and chemical thermodynamic

geothermometer [3]. However, through the application in practice, the geothermal reservoir temperature estimated directly using conventional geothermometers is often far from the actual temperature, and the application is poor. D'Amore et al. calculated the geothermal reservoir temperature of the Zhangzhou geothermal field directly with a cationic chemical geothermometer but failed to obtain consistent results [4]; later, Pang et al. proposed that a more reliable geothermal reservoir temperature estimation could be obtained with an $SiO_2$ mixing model [5]. Wang et al. evaluated the geothermal reservoir temperature in the Tibetan geothermal anomaly using four geothermometers, including $SiO_2$, Na/K, K/Mg, and Na-K-Ca, which yielded mean geothermal reservoir temperatures of 122 °C, 99 °C, 205 °C, and 191 °C, respectively, with significant differences [6]. Later, they determined the most likely range of geothermal reservoir temperatures in the study area based on the principles proposed by Dong et al. and Zhao et al. [7,8]. Guo et al. used $SiO_2$, Na-K-Ca, and K/Mg geothermometers for geothermal reservoir temperature estimation based on the temperature of the Jueyong hot spring in Changdu, Tibet, respectively, and the resulting range of geothermal reservoir temperature was 111–166 °C, with a significant difference [9]. Then, they selected a mixing model for geothermal reservoir temperature estimation through comprehensive analysis. An essential prerequisite for using geochemical geothermometers is that the solute or gas used as a geothermometer should reach an equilibrium state with the minerals in the geothermal reservoir [10]. Furthermore, the content of fluid chemical components may change due to boiling, steam escape, or the mixing of hot water from deep geothermal reservoirs with shallow cold water, which may disrupt the original high-temperature equilibrium environment. Various empirical formulas of geothermometers have been established using the equilibrium reaction of the corresponding components in an aqueous solution with temperature in recent decades [11]. Therefore, the equilibrium state of hot water and minerals must be carefully studied to select a reliable geothermometer.

The southeastern coast of Fujian Province is the main distribution area of medium-low temperature hydrothermal systems in China, and many studies on their temperature fields have been conducted here. Pang et al. calculated the geothermal reservoir temperature of the Zhangzhou geothermal field using the $SiO_2$ mixing model, and the result was 140 °C [5]. Gan et al. integrated the $SiO_2$-enthalpy mixing model with the Na-K-Mg equilibrium analysis and concluded that the geothermal reservoir temperature in the Zhangzhou area is between 120 and 160 °C [12]. Zeng et al. simulated the controlling effect of the fracture system on the temperature field in the Zhangzhou basin. The results showed that the temperature can exceed 130 °C at 2500 m near the fracture system [13]. Recently, a series of Hot Dry Rock exploration holes have been constructed in this area. The temperature of the HDR1 borehole located in Qingquan Forest, Zhangzhou, is only 109 °C at a depth of 4000 m at the bottom of the borehole [14], which differs significantly from the geothermal reservoir temperature estimated by previous authors. Therefore, it is necessary to further explore the geothermal reservoir temperature estimation method in this area. In this paper, taking the typical geothermal system in the Xiamen island–Zhangzhou city area of southeastern China as an example, the integrated water–rock equilibrium analysis, $SiO_2$ geothermometer, multi-mineral equilibrium simulation, and Si-enthalpy mixing model were used to make a quantitative and qualitative analysis of the chemical equilibrium of minerals and fluids in the geothermal system. In addition, the applicability of different methods was determined and then used as a suitable geothermometer to estimate the geothermal reservoir temperature. We hope this study can provide a reference for selecting deep geothermal reservoir temperature estimation methods and for future geothermal exploration in the study area.

## 2. Study Area

The Xiamen island–Zhangzhou city area is located at the front of the Pacific Rim geotectonic activity zone [15]. Since the Mesozoic Yanshanian period movement, many periods of intense magmatic activities and tectonic movements have occurred in the area,

and the geological formations in the area are diverse and complex [16,17]. The hot springs in the study area all belong to the low-medium temperature convective hydrothermal systems [18]. The north-east and north-west fractures constitute the main tectonic framework in this area, and these different fractures, together, form a complete thermal control tectonic system which controls the underground hot water distribution and exposure in the area [19–21]. Spatially, hot springs in the area are mainly distributed on the north-west side of the north-east-oriented Pingtan–Dongshan Fault and the south-east side of the Zhenghe–Nanjing Fault Zone. Generally, the zoning boundary of hot springs is comparable to that of large tectonic lines, and the outcrops of hot springs are not directly related to large tectonic structures—the primary fracture—but are more closely related to the next level of fracture structures [12]. However, from the analysis of individual hot springs, they are mainly exposed at the intersection of the NE- and NW-oriented tectonics. The NW-oriented tectonics in the work area are the newest tectonics in the region, with relatively good underground connectivity and water-richness, providing the main source of recharge and runoff channels for underground hot water, indicating that NW-oriented tectonics are the main water-controlling tectonics in the area. On the other hand, NE-oriented fractures are heat-controlling tectonics [22,23]. In terms of the topography and stratigraphic lithology of the hot springs, most of the hot water in the area is naturally exposed, mainly in the Quaternary alluvial and alluvial marine deposits, except in Xiamen and Zhangzhou city. In contrast, hot springs in volcanic and sedimentary rocks have intrusive rock outcrops in their vicinity [24,25]. Figure 1 shows the study area's location and the hot springs' distribution.

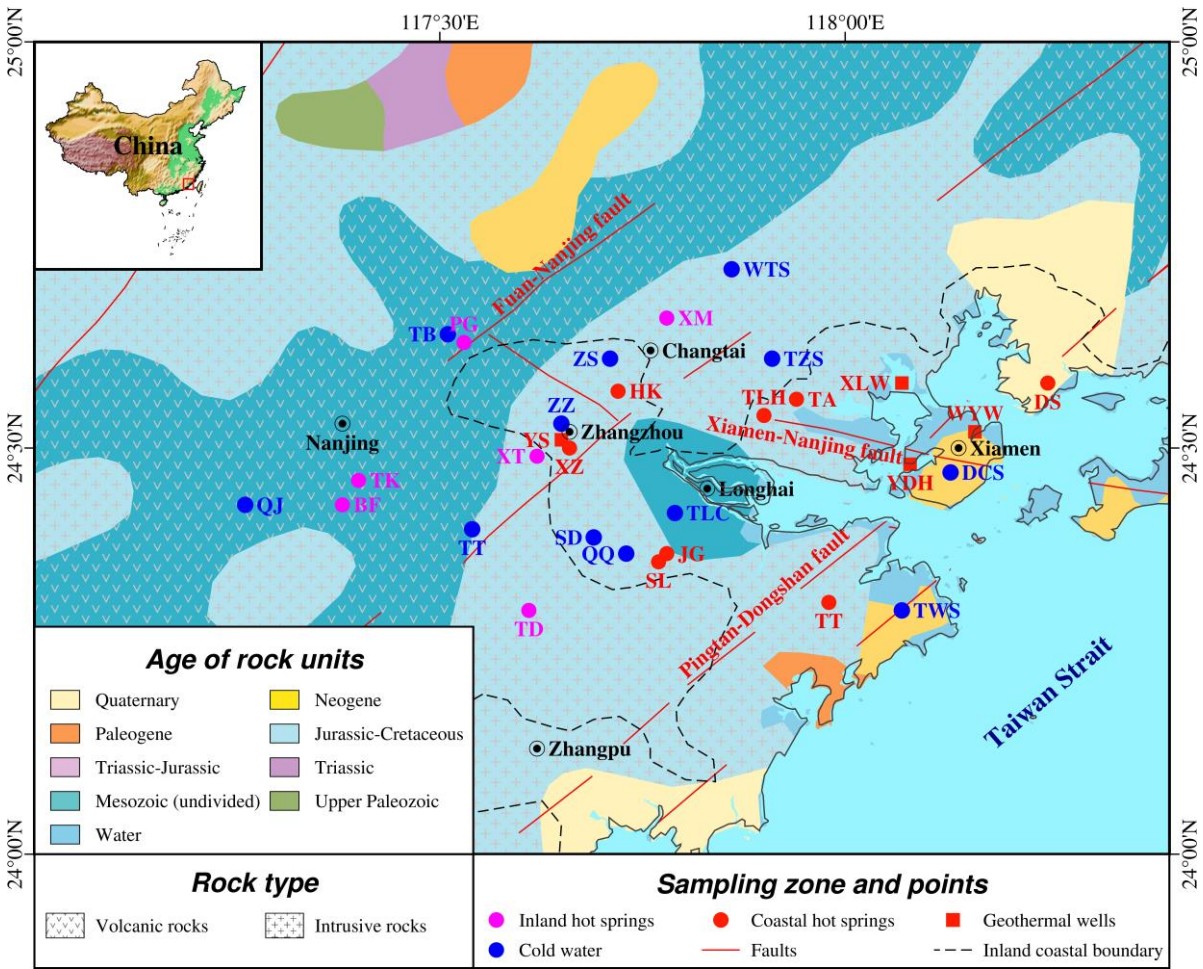

**Figure 1.** Geological map of the study area and the distribution of sampling points.

## 3. Materials and Methods

The typical geothermal systems in the Xiamen and Zhangzhou areas of Fujian Province were selected. Eighteen groups of hot water (hot springs and geothermal wells), twelve groups of cold water (cold springs and shallow groundwater), and one group of rainwater samples were collected to conduct the testing work. Four of the twelve hot water samples were taken from geothermal wells, among which the hot water of Yuandanghu (YDH) was found by late drilling (well depth 202.03 m), and the remaining three were exposed to natural hot springs in the early days. Then, after the hot springs disappeared, people constructed geothermal wells in situ, which were generally less than 200 m deep. The testing items included a complete analysis and trace elements conducted at the Key Laboratory of Groundwater Science and Engineering, Ministry of Natural Resources. Water temperature, pH, and conductivity were tested on-site using a portable water quality analyzer (WTW Multi 3400i). Water samples were first collected by filtration with a 0.45 μm microporous membrane and then stored in polyethylene bottles that were rinsed twice with deionized water and dried. The water was tested according to the National Standard of the People's Republic of China for Natural Mineral Water for Drinking (GB/T8538-2008). Cations using the ICP method using ICAP-6300 for ion concentration detection, anions using the ion chromatograph ICS-1100 for analysis, and the anion and cation balance error are controlled within 3%, and the test results are shown in Table 1.

**Table 1.** Hydrochemical characteristics of the sampling points.

| ID | Types | T (°C) | $K^+$ | $Na^+$ | $Ca^{2+}$ | $Mg^{2+}$ | $Cl^-$ | $SO_4^2$ | $HCO_3^-$ | $HSiO_3^-$ | $\delta^2H$ | $\delta^{18}O$ |
|---|---|---|---|---|---|---|---|---|---|---|---|---|
| Huangkeng(HK) | H | 58.0 | 39.35 | 689.0 | 207.5 | 2.84 | 1216 | 178.2 | 139.0 | 151.30 | −42 | −6.6 |
| Tangdou (TD) | H | 55.6 | 3.29 | 137.6 | 4.04 | 0.04 | 29.88 | 78.34 | 154.1 | 99.84 | −45 | −6.9 |
| Xintang (XT) | H | 47.5 | 3.98 | 185.0 | 24.22 | 0.08 | 135.3 | 159.4 | 90.67 | 82.94 | −47 | −7.2 |
| Tangkeng (TK) | H | 70.5 | 3.05 | 142.0 | 4.73 | 0.04 | 29.88 | 130.0 | 114.8 | 95.42 | −48 | −7.5 |
| Xiazhuang (XZ) | H | 65.1 | 51.14 | 1737 | 1332 | 7.82 | 4868 | 241.9 | 30.22 | 98.28 | −38 | −5.9 |
| Songling (SL) | H | 77.4 | 65.84 | 1746 | 1258 | 11.58 | 4745 | 267.1 | 48.35 | 133.30 | −39 | −5.9 |
| Jingu (JG) | H | 64.5 | 75.06 | 2276 | 1846 | 9.48 | 6415 | 307.4 | 42.31 | 120.90 | −36 | −5.7 |
| Tangtou (TT) | H | 62.7 | 36.45 | 1289 | 399.1 | 1.26 | 2566 | 71.57 | 33.24 | 105.82 | −40 | −5.8 |
| Pangu (PG) | H | 36.5 | 1.69 | 91.57 | 7.22 | 0.13 | 12.30 | 67.68 | 126.9 | 67.73 | −46 | −6.6 |
| Xuemei (XM) | H | 45.8 | 2.57 | 115.0 | 11.94 | <0.013 | 12.30 | 151.4 | 76.16 | 92.82 | −45 | −6.3 |
| Tianli (TLH) | H | 50.5 | 45.82 | 1350 | 985.0 | 4.50 | 3691 | 213.3 | 72.53 | 115.10 | −49 | −6.7 |
| Tangan (TA) | H | 74.4 | 44.06 | 945.6 | 537.4 | 2.19 | 2249 | 220.0 | 36.87 | 132.00 | −51 | −7.3 |
| Baofen (BF) | H | 68.6 | 4.33 | 125.6 | 5.86 | 0.10 | 19.33 | 80.75 | 151.1 | 127.40 | −44 | −6.7 |
| Dongshan (DS) | H | 71.0 | 204.3 | 4916.0 | 2943.0 | 56.60 | 12,214.0 | 347.0 | 48.35 | 118.30 | −26 | −4 |
| Yuandanhu (YDH) | G | 41.0 | 79.50 | 4018.0 | 2882.0 | 163.6 | 10,720.0 | 662.1 | 47.75 | 86.58 | −30 | −4.5 |
| Wuyuanwan (WYW) | G | 56.0 | 130.7 | 4572.0 | 2022.0 | 117.0 | 10,544.0 | 467.5 | 78.58 | 104.00 | −29 | −4.4 |
| Xinlinwan (XLW) | G | 72.0 | 91.12 | 1607.0 | 1029.0 | 5.77 | 4306.0 | 147.9 | 105.8 | 142.50 | −30 | −4.4 |
| Yuanshan (YS) | G | 73.0 | 89.94 | 1598.0 | 1024.0 | 5.64 | 4306.0 | 148.0 | 105.8 | 142.20 | −38 | −5.9 |
| Qingquan forest (QQ) | C | 24.4 | 3.15 | 10.18 | 33.46 | 1.66 | 5.27 | 13.80 | 114.8 | 78.52 | −45 | −7 |
| Shuangdi farm (SD) | C | 24.7 | 2.66 | 6.40 | 8.68 | 0.70 | 2.11 | 3.21 | 36.27 | 37.39 | −44 | −6.8 |
| Taiwushan (TWS) | C | 23.7 | 1.94 | 20.38 | 6.13 | 1.27 | 13.36 | 3.67 | 51.38 | 78.65 | −44 | −6.7 |
| Tianzhushan (TZS) | C | 19.7 | 2.14 | 2.38 | 1.56 | 0.23 | 3.51 | 2.63 | 12.09 | 19.50 | −46 | −6.5 |
| Dacuoshan (DCS) | C | 24.3 | 3.49 | 65.55 | 95.98 | 20.28 | 65.38 | 135.2 | 229.7 | 48.88 | −38 | −5.9 |
| Tatan (TT) | C | 20.1 | 1.16 | 9.41 | 4.34 | 0.81 | 1.76 | 2.09 | 36.27 | 59.28 | −48 | −7.5 |
| Qianjin (QJ) | C | 23.4 | 3.74 | 4.70 | 4.71 | 0.89 | 1.76 | 2.58 | 12.09 | 36.87 | −46 | −7.3 |
| Tianli (TLC) | C | 23.6 | 2.46 | 3.49 | 1.26 | 0.35 | 2.11 | 5.85 | 12.09 | 28.50 | −42 | −6.7 |
| Tianbaobeishan (TB) | C | 24.3 | 1.79 | 7.67 | 11.45 | 3.34 | 5.98 | 3.62 | 48.35 | 39.26 | −41 | −6.6 |
| Wutianshan (WTS) | C | 19.9 | 0.60 | 1.81 | 1.52 | 0.63 | 2.46 | 1.60 | 12.09 | 13.26 | −46 | −7.2 |
| Zhangzhou (ZZ) | C | 25.6 | 8.76 | 21.38 | 17.24 | 5.14 | 33.39 | 6.43 | 36.27 | 50.23 | −41 | −6.5 |
| Zhangshan (ZS) | C | 25.1 | 2.99 | 11.45 | 9.49 | 2.85 | 5.27 | 3.07 | 66.49 | 66.82 | −40 | −6.3 |
| Dongsi(DS) | R | – | 0.85 | 0.99 | 2.39 | 0.16 | 4.57 | 3.82 | 3.02 | 1.20 | −28 | −4.6 |

Notes: for types: H—hot springs; G—geothermal wells; C—cold water; R—rain water.

The data analysis methods mainly include a water–rock equilibrium analysis, multi-mineral saturation index (SI), silica-enthalpy mixing model, and geochemical geothermometer, as detailed in the results and discussion section.

## 4. Results

### *4.1. Hydrochemical Characteristics of Geothermal Fluids*

#### 4.1.1. General Water Chemistry

The characteristics of the hydrochemical distribution of geothermal fluids mainly depend on the composition of the surrounding rock and the geological environment during the geothermal fluid circulation [26–28]. The groundwater temperature plays a significant role in the dissolution and filtration of geothermal fluids on the surrounding rocks. The total dissolved solids (TDS) and the aggregation of some elements in geothermal fluids increase with the temperature increase. In addition, geothermal fluids often form different types of groundwater by mixing with the groundwater of another genesis during their participation in the natural water cycle. According to the geological environment of the study area, the geothermal system of the study area can be divided into two subzones, the coastal zone and the inland zone, which are used to analyze their different geothermal fluid chemical characteristics. There are 12 geothermal sites in the coastal zone of the study area. Due to the dissolution of residual salts in the marine sediments or the sequestered ancient seawater or the mixing of part of the seawater recharge through the fractures connected with seawater, the total dissolved solids (TDS) of the geothermal fluid in the area is high, ranging from 2.5 to 20.85 g/L, which is brackish-saline water. The anions in the fluid are mainly $Cl^-$, the cations are mainly $Na^+$ and $Ca^{2+}$, and the water quality types include Cl-Na-Ca, Cl-Ca-Na, and Cl-Na. There are six hot springs in the inland area, mainly located in mountainous basins, depressions, and river valleys in the hilly mountainous region. The geothermal fluid is mainly recharged by atmospheric precipitation and eventually emerges to the surface after deep circulation through fractures. The chemical water components in the geothermal fluid in this area mainly come from the dissolution and filtration of the geothermal fluid on the surrounding rocks, the water cycle alternates slowly, and the total dissolved solids (TDS) is low, ranging from 0.31 to 0.63 g/L, all of which are freshwater. The water quality type is closely related to the composition of the surrounding rocks and water temperature. The anion is mainly $HCO_3^-$, $SO_4^{2-}$, and the cation is $Na^+$, $Ca^{2+}$. The water quality type is diverse, mainly including $HCO_3$-Na, $HCO_3$-Ca, and $HCO_3$-$SO_4$-Na -Ca, $SO_4$-Cl -Na (Figure 2).

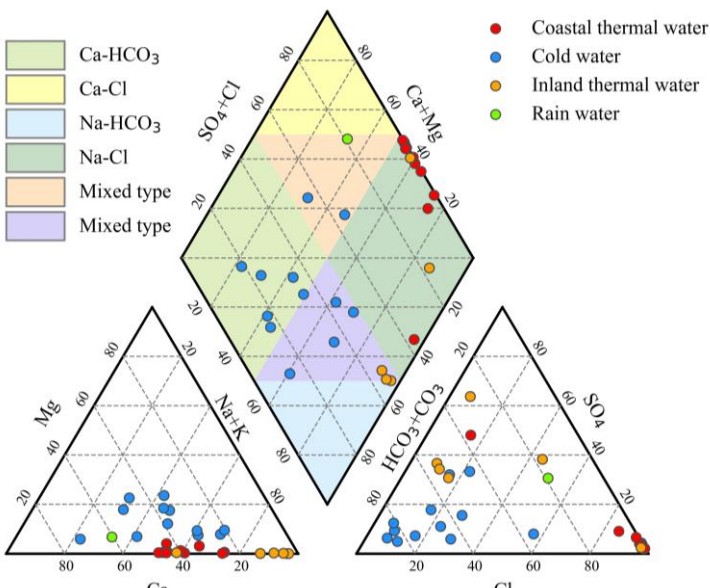

**Figure 2.** Piper diagram of different water bodies in southeastern China.

4.1.2. Stable Isotopic Characteristics

The $\delta^2$H ranges of cold springs and groundwater wells are −48‰~−38‰ and −45‰~−40‰, respectively, and the $\delta^{18}$O ranges are −7.5‰~−5.9‰ and −7‰~−6.3‰, respectively. The groundwater wells in the study area are generally shallow. Both the groundwater and cold springs receive direct recharge from precipitation, so their stable isotope components do not differ much, and they are all located near the GMWL [29] and LMWL [30] (Figure 3). The isotopic values of the thermal water samples also show a limited range of variability: −51‰ to −44‰ and −7.5‰ to −6.5‰ for the inland hot water and −49‰ to −26‰ and −6.7‰ to −4‰ for the coastal hot water, respectively. The hot water is distributed near the GMWL and LMWL, indicating that the hot water receives recharge from local precipitation. No significant oxygen drift in hot water indicates that the geothermal reservoir is not high in temperature and maybe indicates a medium-low temperature geothermal system. The seawater mixing line in the study area can be mapped based on the intersection of GMWL and LMWL and the isotopic composition of seawater ($\delta^2$H$_{seawater}$ = 0, $\delta^{18}$O$_{seawater}$ = 0) (Pang et al., 2017). More than half of the coastal hot water sites fall on or near this line, indicating that the geothermal fluids are influenced by seawater mixing, which is also entirely consistent with the anomalously high Cl$^-$ content in the coastal hot water (Table 1).

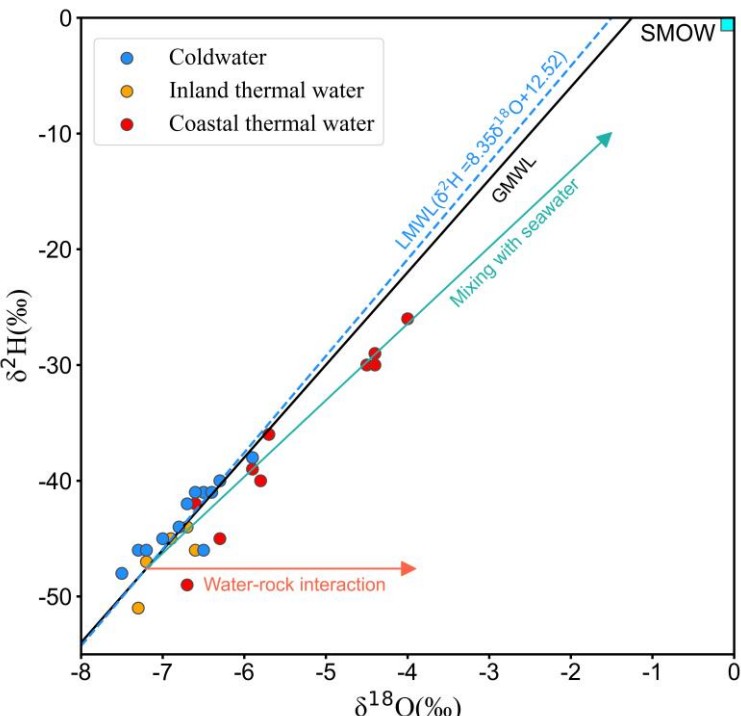

**Figure 3.** $\delta^2$H–$\delta^{18}$O relationships of water samples in southeastern China.

In addition, Arnorsson et al. suggested that the content of the conservative element Cl$^-$ in groundwater with the presence of mixing tends to show a significant linear relationship with $\delta^2$H and $\delta^{18}$O [2]. The relationship between Cl$^-$ and the hydrogen-oxygen isotopes of hot spring water in the study area is shown in Figure 4, which shows that there is a good correlation between Cl$^-$ and $\delta^2$H as well as $\delta^{18}$O in hot water in the coastal zone, while there is no apparent correlation in the inland zone, which further confirms the existence of mixing between geothermal systems and seawater in the coastal zone.

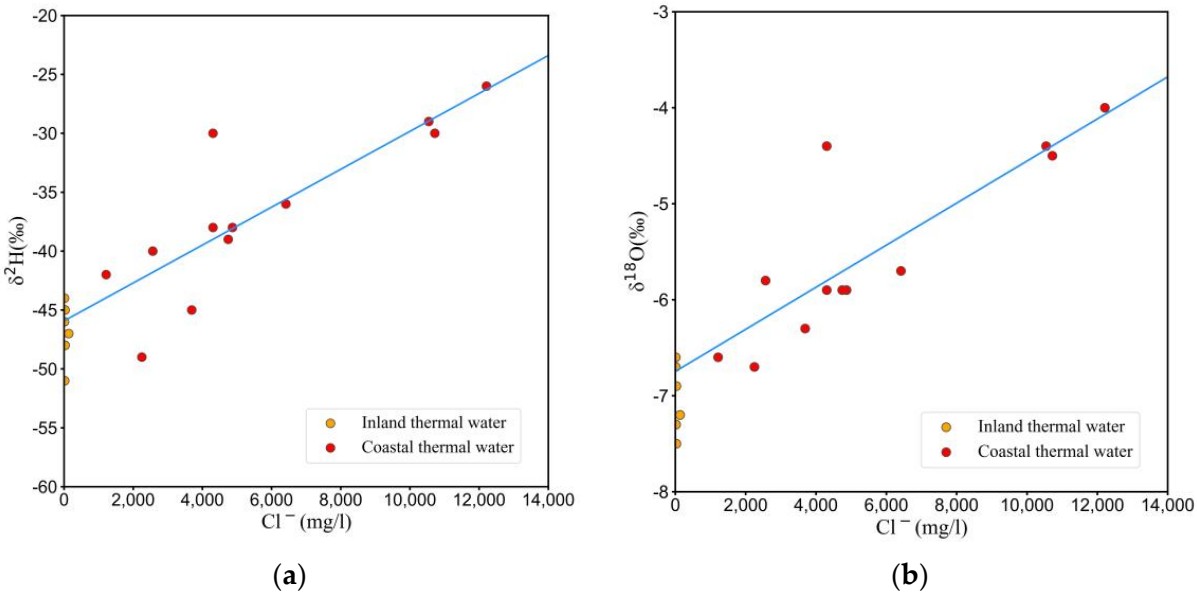

**Figure 4.** $Cl^-$-$\delta^2H$ (**a**), $Cl^-$-$\delta^{18}O$ (**b**); correlation of geothermal water in southeastern China.

### 4.2. Water–Rock Balance Analysis

#### 4.2.1. Na-K-Mg Geoindicator

The Na-K-Mg triangle diagram, proposed by Giggenbach [31], is divided into three regions: complete equilibrium, partial equilibrium, and immature water, and it is often used to evaluate the state of the water–rock equilibrium and to distinguish different types of water samples. The advantage of this method is that the equilibrium state of a large number of water samples can be judged simultaneously on the same diagram, and the mixed water and equilibrium water can be well separated. The Na, K, and Mg contents of various water bodies in the study area were linearly transformed and projected onto the Na-K-Mg equilibrium triangle (Figure 5), in which all shallow groundwater and cold springs were located in the immature water region at the right lower apex of Mg ions, indicating that they were still in the primary stage of water–rock interaction, while all samples of hot springs and geothermal wells were in the partial equilibrium or mixed water region, reflecting that the ionic equilibrium between the water and rock has not yet been reached, and dissolution is still ongoing, or the hot water was mixed with cold water. In addition, the geothermal reservoir temperatures of most of the hot springs in the Na-K-Mg equilibrium triangle diagram fall between 120 °C and 160 °C. The geothermal reservoir temperatures of the Tangan hot spring all fall between 160 °C and 180 °C, and the geothermal reservoir temperatures of hot water from the Huangkeng hot spring, Henglinwan geothermal well, and Yuanshan geothermal well all fall between 180 °C and 200 °C.

Overall, the cationic geothermometer is unsuitable for most of the water samples in the study area and will not yield reasonable geothermal reservoir temperatures. However, it has some reference value for samples in the equilibrium area or on the equilibrium line.

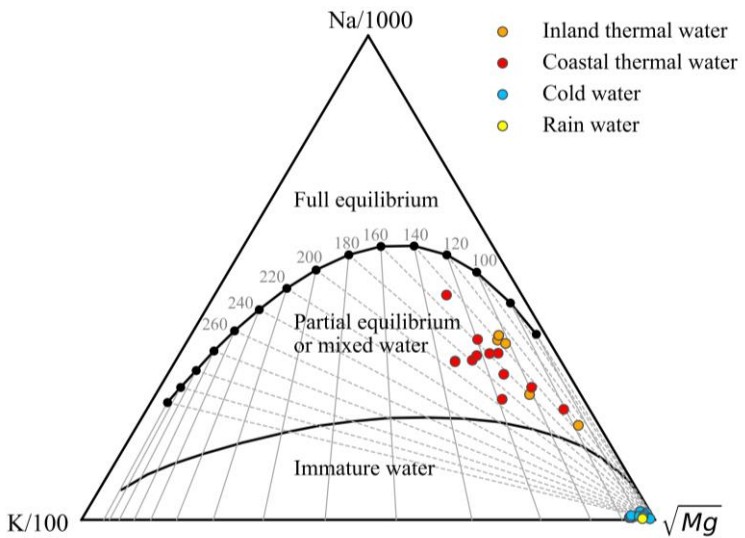

**Figure 5.** Na-K-Mg diagrams for various water in southeastern China.

4.2.2. SiO$_2$, Na/K, and 1000/T Diagrams

From Figure 6a, it can be seen that most of the hot water sample points in the study area fall between the theoretical equilibrium dissolution line of Cristobalite and Opalite, indicating that the samples are saturated or supersaturated with quartz and chalcedony. In addition, most of the sample points in the study area fall below the Na/K equilibrium line. Only the water samples from the Tangkeng hot spring are located between the Na/K equilibrium line [32] and the Na/K equilibrium line [33] (Figure 6b), indicating that most of the water samples may be influenced by cold water mixing. The water–rock interaction has not reached full equilibrium.

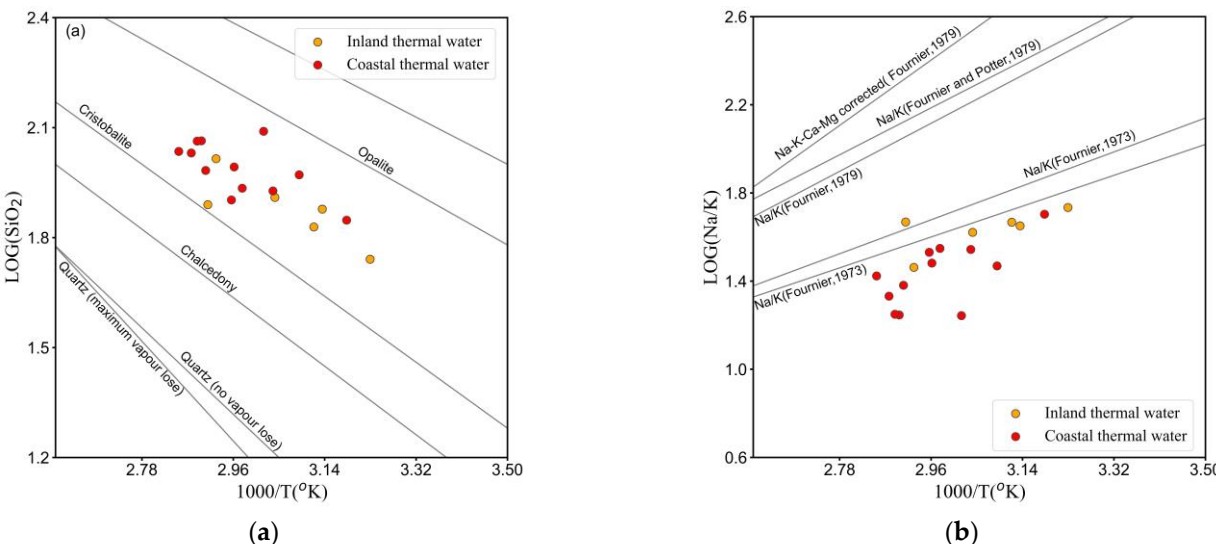

**Figure 6.** Plots of SiO$_2$ (**a**), Na/K (**b**), and 1000/T for geothermal water in southeastern China.

Therefore, from the Na-K-Mg triangle diagram and log (Na/K)-1000/T diagram, it can be determined that most of the samples in the study area were influenced by the mixing effect, and the cationic geothermometer is not suitable, but there is a particular reference value. At the same time, some scholars proposed that the mixing process of cold and hot water has little effect on the Na/K value, and an Na/K geothermometer can respond to the temperature of deep, hot water. Therefore, the Na-K-Ca geothermometer can be used for the hot spring samples in Tangkeng, which are above the Na/K equilibrium line. Based on the log (SiO$_2$)-1000/T diagram, it can be determined that the SiO$_2$ (quartz or chalcedony)

geothermometer can be used for the study area samples, which can usually reflect the geothermal reservoir temperature better after the mixing effect occurs.

The results of the SiO$_2$ geothermometer for each hot spring in the study area are shown in Table 2.

**Table 2.** Geothermal reservoir temperature for different geothermal systems in the study area by the SiO$_2$ geothermometer.

| Sampling Location | Chalcedony—No Vapor Loss | Chalcedony—Maximum Vapor Loss | Quartz—No Vapor Loss | Quartz—Maximum Vapor Loss |
|---|---|---|---|---|
| Huangkeng | 123.8 | 119.4 | 149.1 | 142.7 |
| Tangdou | 98.0 | 98.6 | 125.9 | 123.1 |
| Xintang | 87.6 | 90.0 | 116.3 | 115.0 |
| Tangkeng | 95.4 | 96.4 | 123.5 | 121.1 |
| Xiazhuang | 97.1 | 97.8 | 125.0 | 122.4 |
| Songling | 115.5 | 112.8 | 141.7 | 136.5 |
| Jingu | 109.4 | 107.9 | 136.2 | 131.9 |
| Tangtou | 101.4 | 101.3 | 129.0 | 125.8 |
| Pangu | 76.8 | 81.0 | 106.4 | 106.5 |
| Xuemei | 93.8 | 95.1 | 122.1 | 119.9 |
| Tianli | 106.4 | 105.4 | 133.5 | 129.6 |
| Tangan | 114.9 | 112.3 | 141.2 | 136.1 |
| Baofen | 112.7 | 110.5 | 139.2 | 134.4 |
| Yuandanhu | 89.9 | 91.9 | 118.5 | 116.9 |
| Wuyuanwan | 100.4 | 100.5 | 128.0 | 125.0 |
| Xinglingwan | 119.8 | 116.2 | 145.6 | 139.8 |
| Dongshan | 108.1 | 106.8 | 135.0 | 130.9 |
| Yuanshan | 119.7 | 116.1 | 145.5 | 139.7 |

*4.3. Multiple Mineral Equilibrium Approaches*

Reed and Spycher proposed the multi-mineral equilibrium method to determine the chemical equilibrium state in geothermal systems [34]. The principle is to treat the dissolution state of multiple minerals in water as a function of temperature. Suppose a group of minerals approaches equilibrium at a particular temperature simultaneously. In that case, the hot water has reached the equilibrium with this group of minerals, and the temperature at the equilibrium is the deep geothermal reservoir temperature. Neither mixed water nor hot water that has not reached the mineral dissolution equilibrium can bring multiple minerals into the equilibrium at a specific temperature simultaneously. Therefore, according to the multi-mineral equilibrium diagram, we can determine whether hot water is mixed with shallow cold water, whether hot water is in an equilibrium with certain minerals, and whether the temperature corresponds to the equilibrium.

The degree of saturation of each mineral can be judged according to the saturation index SI of the minerals in geothermal water. The rules are as follows: SI > 0, indicating supersaturation; SI = 0, indicating saturation; SI < 0, indicating unsaturation. The saturation index is defined as follows,

$$SI = \log(Q/K), \tag{1}$$

where K is the solubility of minerals in geothermal water, mol/L; Q is the actual ionic activity product of minerals dissolved in geothermal water, mol/L.

The concentration of aluminum ions in the geothermal water in the study area was below the detection limit. Therefore, it was not detected, thus preventing the production of aluminum-bearing silicate minerals. One or two aluminum-bearing silicate minerals have reached the equilibrium in most geothermal systems. Accordingly, Pang and Reed pioneered the use of immobilized aluminum to restore the equilibrium state of aluminum-bearing silicate minerals in geothermal systems [35]. Geothermal systems generally contain multiple components and are in a non-homogeneous chemical equilibrium. The equilibrium

of aluminosilicate minerals is interdependent, as all minerals contain specific chemical components. When the Al concentration is unknown, the Al activity can be estimated at different temperatures of interest by assuming that the Al activity is fixed by some Al-bearing mineral such as microcline feldspar. The Al concentration values are obtained and then used to calculate the Q/K values of other aluminosilicate minerals. This method of estimating Al concentration values may be valid for given geothermal water since most geothermal water is in equilibrium with at least two aluminosilicate minerals.

The equilibrium state of hot water in the study area was analyzed by fixing Al with microplagioclase. Based on the geothermal fluid water quality data, the PHREEQCI program can calculate the dissolved minerals' activity coefficients, simulate the chemical composition and mineral formation in the geothermal fluid, and further simulate the saturation index SI of dissolved minerals in hot water. We first calculated the saturation index SI values of various minerals at different temperatures using the PHREEQCI program. We then plotted the SI- T with the temperature as the horizontal coordinate and the log (Q/K) value as the vertical coordinate curve. When different minerals reach saturation at the same temperature (curves' intersection with SI = 0), it is the temperature of the geothermal reservoir. Based on the geological survey data of the study area, we selected 11 common minerals for simulation: fluorite, chalcedony, quartz, petalite, tridymite, K-feldspar, illite, and muscovite.

The SI-T curves for all hot water samples in the study area are shown in Figure 7. As can be seen in Figure 7, the dissolution curves of minerals in most of the hot spring samples are well converged, except for the poorly converged dissolution curves of different minerals in the Tangkeng (Figure 7d) and Pangu (Figure 7i) hot spring samples. The corresponding horizontal coordinate is the geothermal reservoir temperature of this geothermal system. The geothermal reservoir temperatures of most of the geothermal systems in the study area range from 125 to 160 °C, which is in general agreement with the results of the $SiO_2$ geothermometer (without vapor loss) (Table 2).

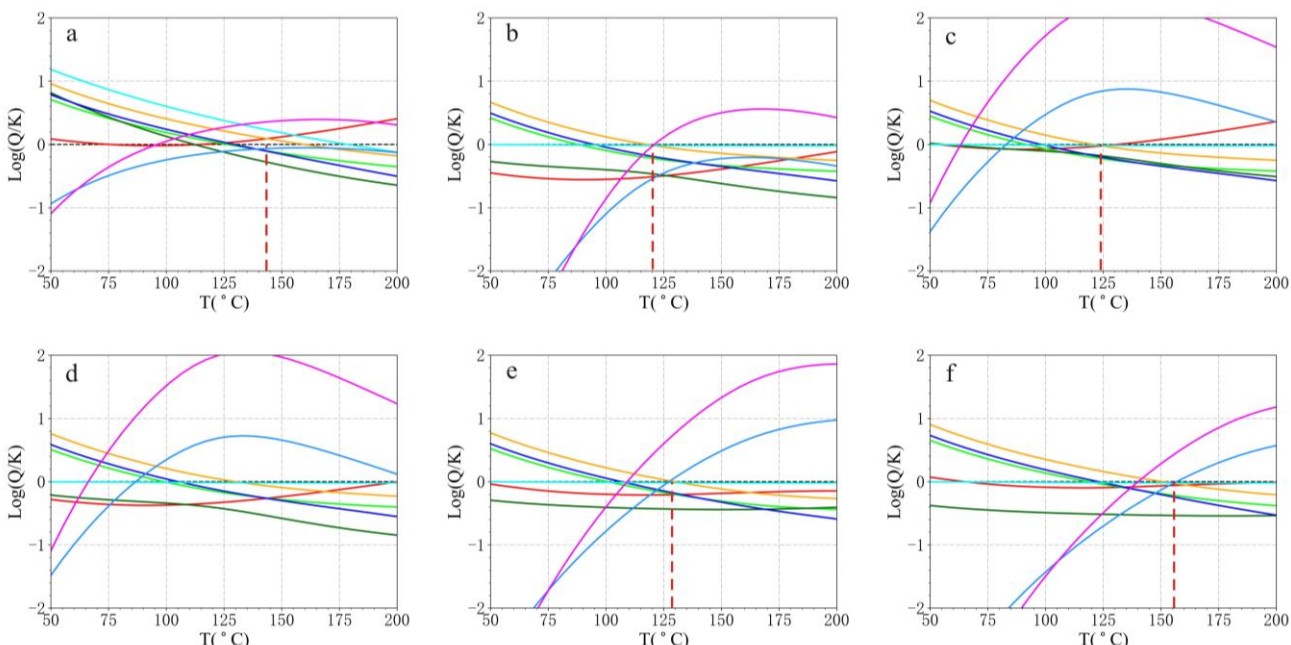

**Figure 7.** *Cont.*

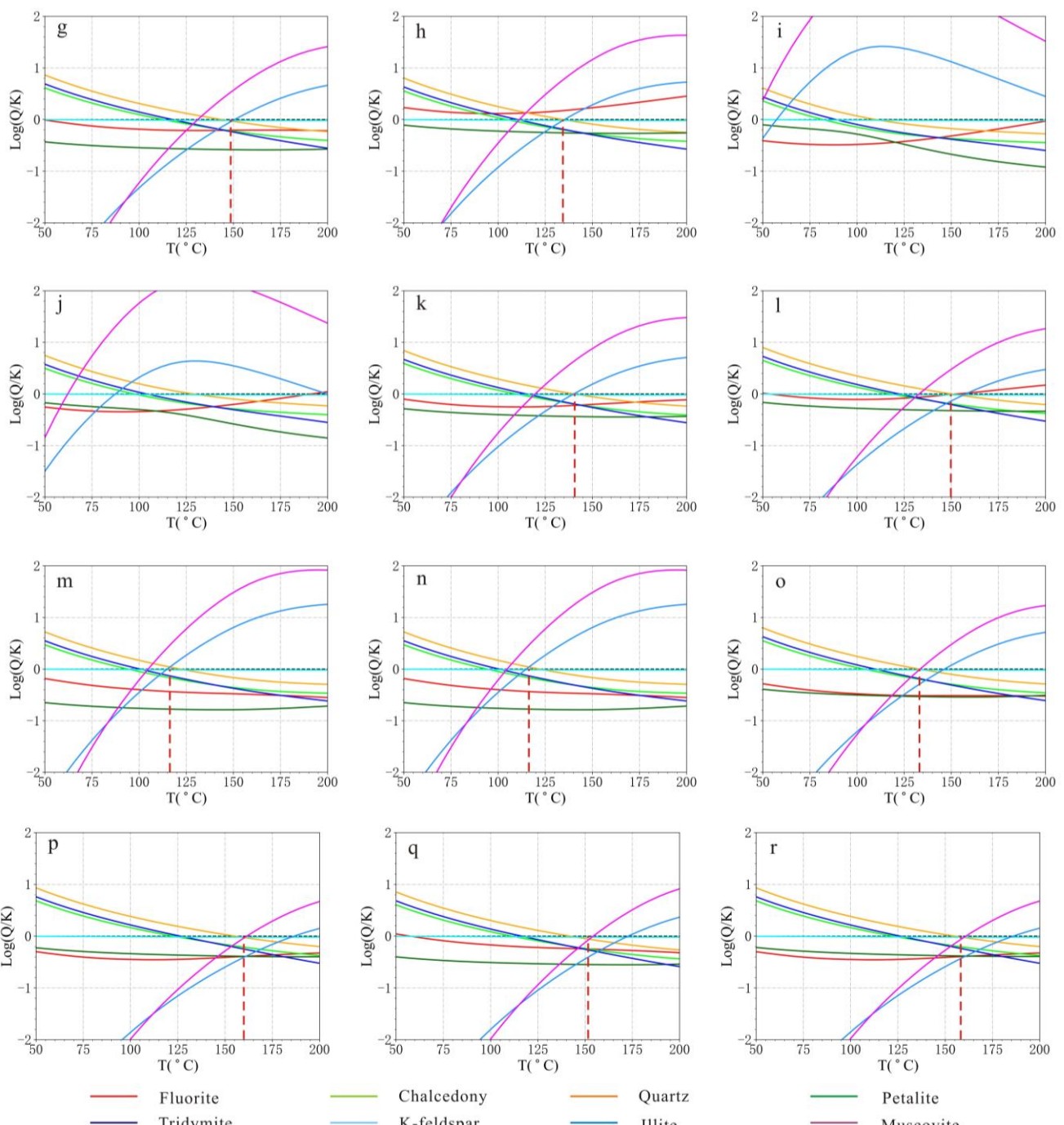

**Figure 7.** Mineral saturation diagram for the typical hot spring in southeastern China: (**a**)—Huangkeng; (**b**)—Tangdou; (**c**)—Xintang; (**d**)—Tangkeng; (**e**)—Xiazhuang; (**f**)—Songling; (**g**)—Jingu; (**h**)—Tangtou; (**i**)—Pangu; (**j**)—Xuemei; (**k**)—Tianli; (**l**)—Tangan; (**m**)—Baofen; (**n**)—Yuandanhu; (**o**)—Wuyuanwan; (**p**)—Xinlinwan; (**q**)—Dongshan; (**r**)—Yuanshan.

### 4.4. Silicon-Enthalpy Mixing Model

The SI-enthalpy mixing model can be used to estimate the geothermal reservoir temperature [32]. The mixing model is based on mixing the enthalpy of hot and cold water and the $SiO_2$ content in different proportions, resulting in the final state of the enthalpy and $SiO_2$ content of the hot water when it is exposed.

The model equation is:

$$S_c X_1 + S_h(1 - X_1) = S_s \tag{2}$$

$$SiO_{2c}X_2 + SiO_{2h}X_2(1 - X_2) = SiO_{2s} \qquad (3)$$

where $S_c$ is the enthalpy of cold water (J/g) and $S_s$ is the final enthalpy of hot spring water (J/g). The enthalpy of saturated water below 100 °C equals the Celsius temperature of the water. When the water temperature exceeds 100 °C, the relationship between the temperature and enthalpy of the saturated water can be obtained in the relevant data sheet. $S_h$ is the initial enthalpy of hot water (J/g); $SiO_{2c}$, $SiO_{2h}$, and $SiO_{2s}$ represent the mass concentration of $SiO_2$ of cold water, hot water, and hot spring water (mg/L); X is the mixing ratio of cold water.

The cold water temperature is taken as the average temperature of the cold spring measured in the field, 21 °C, and the cold water $SiO_2$ content is taken as the average of the cold spring content. The initial temperature of hot water is assumed to be 50–300 °C, and the corresponding enthalpy value can be found in the relevant table. The temperature and $SiO_2$ content of hot spring water was based on the measured values in the field. The enthalpy of each temperature and the mass concentration of $SiO_2$ are substituted into formulas (2) and (3), respectively, to find out the $X_1$ and $X_2$ values at different temperatures and plot the graph of X and hot water temperatures at different temperatures (Figure 8). The intersection of $X_1$ and $X_2$ curves is the calculated thermal storage temperature value. The results are shown in Table 3.

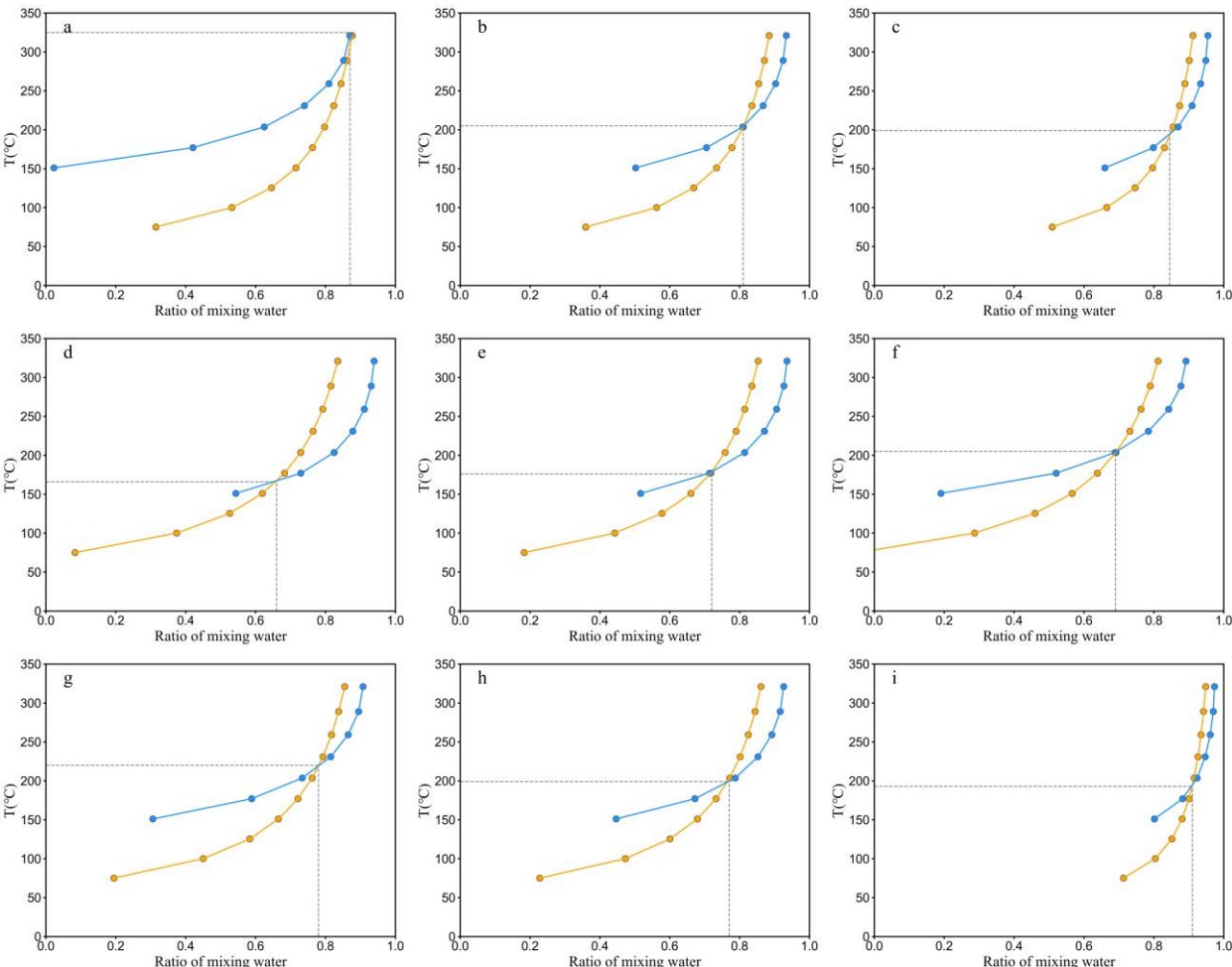

**Figure 8.** *Cont.*

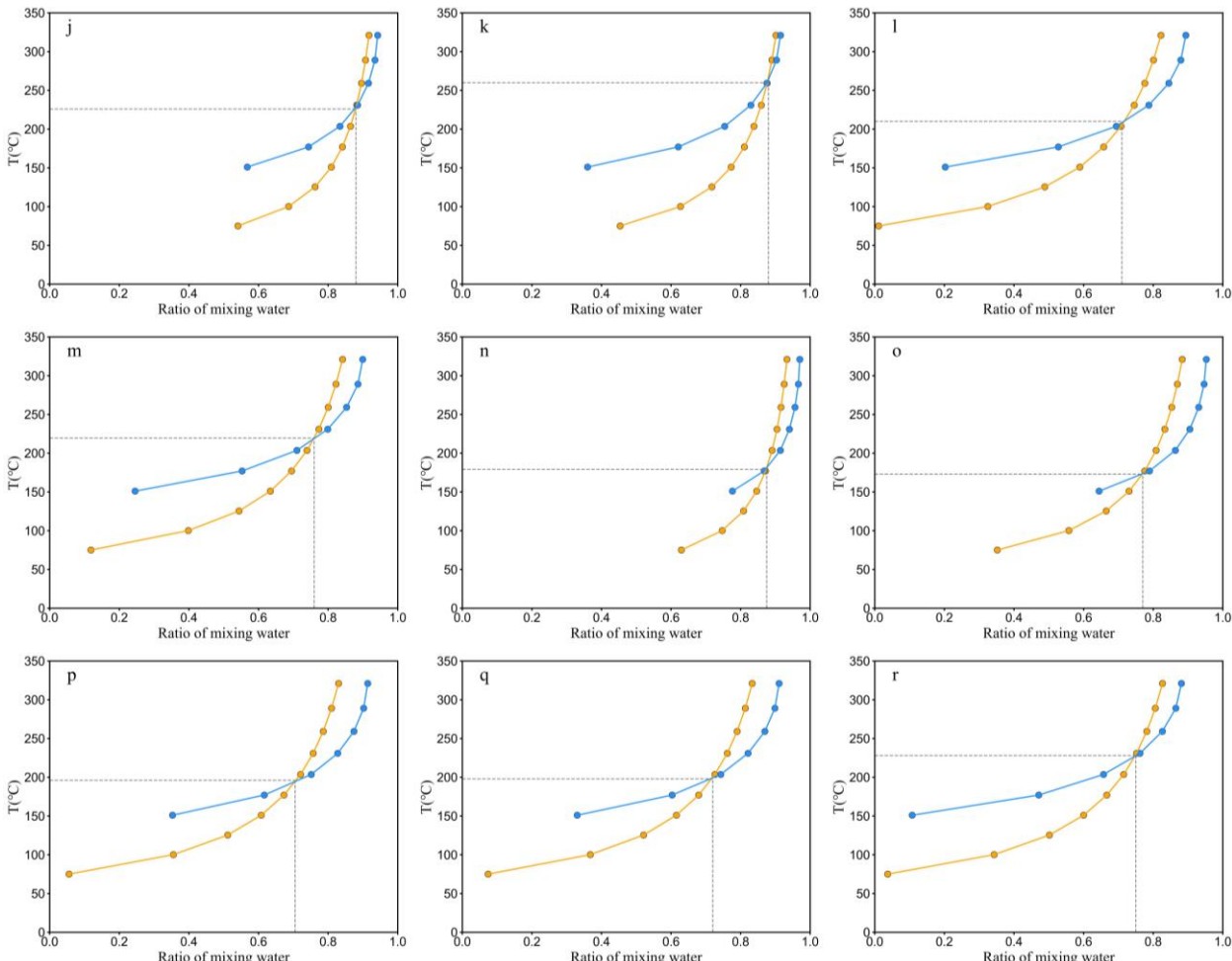

**Figure 8.** Graphical representation of the ratio of deep hot water temperature to mixed cold water in the study area ((**a**)—Huangkeng; (**b**)—Tangdou; (**c**)—Xintang; (**d**)—Tangkeng; (**e**)—Xiazhuang; (**f**)—Songling; (**g**)—Jingu; (**h**)—Tangtou; (**i**)—Pangu; (**j**)—Xuemei; (**k**)—Tianli; (**l**)—Tangan; (**m**)—Baofen; (**n**)—Yuandanhu; (**o**)—Wuyuanwan; (**p**)—Xinlinwan; (**q**)—Dongshan; (**r**)—Yuanshan).

**Table 3.** Geothermal reservoir temperatures and the proportion of cold water mixed for the hot springs obtained from the mixing model.

| Hot Springs | Geothermal Reservoir Temperature (°C) | The Proportion of Cold Water | Hot Springs | Geothermal Reservoir Temperature (°C) | The Proportion of Cold Water |
|---|---|---|---|---|---|
| Huangkeng | 325 °C | 0.87 | Xuemei | 226 °C | 0.88 |
| Tangdou | 205 °C | 0.81 | Tianli | 260 °C | 0.88 |
| Xintang | 199 °C | 0.85 | Tangan | 210 °C | 0.71 |
| Tangkeng | 166 °C | 0.66 | Baofen | 219 °C | 0.76 |
| Xiazhuang | 176 °C | 0.72 | Yuandanhu | 179 °C | 0.88 |
| Songling | 205 °C | 0.69 | Wuyuanwan | 173 °C | 0.77 |
| Jingu | 220 °C | 0.78 | Xinglinwan | 196 °C | 0.71 |
| Tangtou | 199 °C | 0.77 | Dongshan | 198 °C | 0.72 |
| Pangu | 193 °C | 0.91 | Yuanshan | 228 °C | 0.75 |

The Si-enthalpy mixing model is based on the assumption that the dissolved $SiO_2$ is saturated in the deep hot water and is mixed with cold water without the secondary equilibrium of the water–rock reaction, which is often far from the ideal model in practice. From previous research results, it is known that the results of the Si-enthalpy mixing

model are usually higher than the actual situation. When the Chinese Academy of Sciences used the Si-enthalpy model to estimate the temperature of geothermal water in more than 40 springs or boreholes in the Yangbajing geothermal field, Tibetan Plateau, the results were 60–70 °C higher than those calculated by the adiabatic cooling process or the conduction cooling process [36]. The possible influencing factor is that the hot water may have lost steam due to over-expansion before mixing with cold water, thus making the measured value of $SiO_2$ content in hot water high.

A comparison of these methods leads to the following conclusions (Figure 9): (1) there is a mixture of hot water and seawater or cold water, the geothermal system is in a non-equilibrium state, and the cationic geothermometer is not suitable for geothermal reservoir temperature estimation in the study area; (2) the silica-enthalpy mixing model obtains the highest geothermal reservoir temperature range (166–325 °C), which is inconsistent with the traditional understanding of the medium-low temperature geothermal system in the study area; (3) the results of the quartz geothermometer without vapor loss (106.4–149.1 °C) are closer to the results of the multi-mineral saturation index (SI) (116–160 °C). In general, an accurate prediction of the geothermal reservoir temperature can be achieved by using the saturation index (SI) of multiple minerals at different temperatures. It consists of simulating a gradual increase in hydrothermal fluid temperature and observing the changes in different saturation states of a given mineral under reservoir equilibrium conditions. The temperature estimation of the geothermal reservoir using a silica geothermometer is based on the silica minerals solubility (quartz, chalcedony, and cristobalite), which is controlled by pressure and temperature. Mixing hot water with cold water at shallow depths during ascent causes a decrease in $SiO_2$ concentration; therefore, the temperature calculated with the $SiO_2$ geothermometer is usually considered to be the minimum temperature of the reservoir [37]. Therefore, the results of the quartz geothermometer without vapor loss and the multimineral saturation index (SI) can be selected as the range of geothermal reservoir temperatures for different geothermal systems in the study area.

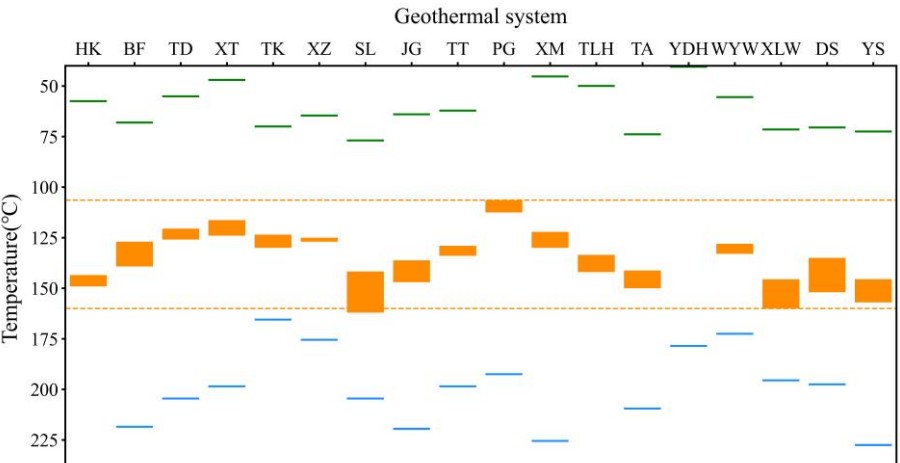

**Figure 9.** Geothermal reservoir temperatures of different geothermal systems in the study area obtained by different methods. The orange bar graph shows the range of geothermal reservoir temperatures determined by the quartz geothermometer (without vapor loss) and the multiple mineral saturation index method; the blue line indicates the geothermal reservoir temperature obtained from the silica-enthalpy mixing model, and the green line shows the hot water temperature at the surface of the corresponding geothermal system.

## 5. Discussions

There are numerous geothermal reservoir temperature estimation methods, but the results obtained by different methods vary greatly, mainly because they all have a specific range of applicability. In some exceptional cases, this range of applicability is exceeded,

resulting in significant deviations in the results. Therefore, studying the applicability of different geothermal reservoir temperature estimation methods is crucial.

In essence, the judgment of the applicability of many geothermal reservoir temperature estimation methods involves three main aspects. First, to judge whether the selected evaluation parameters can objectively reflect the original state of the geothermal reservoir at high temperatures—for example, whether the components change during the fluid transport from the deep strata to the surface due to water–rock action, steam loss, and fluid mixing. Second, to judge whether the problem under study meets the assumptions of the geothermal reservoir temperature estimation method. Third, to discern whether the relationship between the evaluation parameters and temperature will change due to specific circumstances—for example, whether different water types and the presence or amount of certain minerals or ions will affect the thermodynamic equilibrium of the water body.

The cationic geothermometer is mainly based on Na-K, Na-K-Ca, and Na-Li in water. The dissolved components ratio of the exchange reaction depends on the temperature. You can use the ratio of these components as a geothermometer. As the water–rock reaction and mixing between water bodies occurring during the upwelling of hot water will affect the accuracy of the predicted temperature of the cationic geothermometer, it is usually necessary to first use the most commonly used Na-K-Mg triangle diagram method to determine the mixing relationship between geothermal water and cold water and the water–rock equilibrium state and then select a suitable cationic geothermometer to calculate the geothermal reservoir temperature according to its water–rock equilibrium state. If there is no equilibrium, a silica geothermometer is used to circumvent the effect of decreasing hot water temperature. For the cationic geothermometer, from the point of view of the whole system chemical reaction equilibrium, the convergence of the multi-mineral saturation index can be used to determine the equilibrium state of the minerals in the water body and the geothermal reservoir temperature according to the temperature corresponding to the convergence. In the case of cold and hot water mixing, the concentration of minerals and ions in the original hot water has been diluted, and it is difficult to reflect the characteristics of the original hot water. In such a case, the geothermal reservoir temperature can be evaluated by using the Si-enthalpy model, Cl-enthalpy model, and Si-carbonate model, which are cold and hot water mixing models.

No single geothermometer is a panacea. A systematic analysis of the influence of water and rock outside the geothermal system and the disruption of the chemical reaction equilibrium of all fluids is necessary to select the most reasonable geothermal thermometer. Often, the combination of several methods can avoid the limitations of a single method, and several methods estimate a typical temperature interval to represent the actual temperature of the subsurface thermal reservoir.

## 6. Conclusions

(1) The geothermal anomaly area in the study area can be divided into coastal and inland zones. The hot water in the inland zone is dominated by $HCO_3$-Ca type water with a low TDS. On the other hand, the coastal zone is dominated by Cl-Na type water with a high TDS, indicating that the hot water is influenced by seawater recharge or dissolved residual salt in the marine sediment.

(2) The isotopes of hot water samples in the study area are distributed near the GMWL and LMWL, showing a limited range of variability, indicating that the geothermal receives local precipitation recharge. Most of the hot water in the inland area falls below the Na/K equilibrium line, indicating that the hot water may be influenced by cold water mixing. Most of the hot water in the coastal zone falls on the seawater mixing line, and the Cl content shows an apparent linear relationship with $\delta^2 H$ and $\delta^{18}O$, indicating that the geothermal fluid is influenced by the seawater mixing effect. Therefore, using cationic geothermometers to calculate the geothermal reservoir temperature in these areas is unsuitable.

(3) The results of the geothermal reservoir temperature estimation of the geothermal system in the study area using silica geothermometers (without vapor loss) and multiple mineral saturation indices are reliable. However, due to the mixing of hot water with cold water at shallow depths during ascent, the temperature calculated with the $SiO_2$ geothermometer is usually considered the minimum temperature of the reservoir. Therefore, the results of the quartz geothermometer (without vapor loss) and the multimineral saturation index (SI) can be selected as the range of geothermal reservoir temperatures for different geothermal systems in the study area.

**Author Contributions:** Conceptualization, W.L. and X.Y.; methodology, W.L.; software, X.Y.; validation, X.Y.; formal analysis, X.Y.; investigation, W.L.; resources, W.L.; data curation, W.L.; writing—original draft preparation, X.Y.; writing—review and editing, W.L.; visualization, X.Y.; supervision, W.L.; project administration, W.L.; funding acquisition, W.L. All authors have read and agreed to the published version of the manuscript.

**Funding:** This research was supported by the National Key R&D Program of China (Grant No. 2021YFB1507401) and Qinghai Province Clean Energy Special Funds (Grant No. 2022013004qj004).

**Data Availability Statement:** The data used to support the findings of this study are included within the article.

**Conflicts of Interest:** The authors declare no conflict of interest.

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
