# Peer review of "Temperature Estimation of a Deep Geothermal Reservoir Based on Multiple Methods: A Case Study in Southeastern China"

_water, doi:10.3390/w14203205_

Round 1

Author Response

Dear Reviewer:

Thank you very much for reviewing and commenting on our paper, “Temperature Estimation of Deep Geothermal Reservoir Based on Multiple Methods: A Case study in Southeastern China”(Manuscript Number: Water-1906621). Your comments were valuable and helped us revise and improve this paper; they will also help guide our future research. We have revised our manuscript and made corrections to address your concerns. The revised portions of our manuscript are highlighted in blue. The main corrections to the paper and our responses to reviewer comments follow:

General comments:

  1. The paper discussed the interesting topic in the geothermal exploration. Selecting geothermometer the most proper geothermometer method will reduce the exploration risk. Therefore, this topic is important.
  2. The authors described the importance of the geothermometer method to infer the reservoir temperature.
  3. The objective of the study is clearly stated.

Response: Thank you for your comments.

Comment on Introduction:

  1. In the line number 37 – 39 the authors claimed the inaccuracy of conventional geothermometer in predicting geothermal reservoir. However, the authors did not give any example or references. It will be better if the authors can give the case.

Response: Thank you for your comment. The authors have added examples of direct geothermal reservoir temperature calculations using chemical geothermometers in chronological order to the article and explain the option they eventually took. These examples further demonstrate that the immediate use of chemical geothermometers for thermal reservoir temperature calculations is inaccurate.

Comment on Study area:

  1. The authors described that the occurrence of hot springs is related to the fracture or geological structures, however the authors did not provide the geological map showing the geological structure of the studied area. Please provide the geological map, so the reader can imagine the structural control and give information about the lithology of the studied area.

Response: Thank you for your comment. The authors have replaced Figure 1 with a geological map of the study area, which contains information on fractures, lithology, and the distribution and type of sampling points.

  1. The authors described that the area is near coastal area, but the Figure 1 that is the location map did not indicate the coastal line. Please revised Figure 1 with indicate the coastal line so the reader can have idea the distance of the springs from the coastal. Furthermore, the reader may have idea about the contribution of sea water that may mix with thermal water.

Response: Thank you for your comment. In the newly replaced Figure 1, information such as coastal line and names of the adjacent straits have been included.

  1. When we discuss about geothermal water, it is better if we give information about the type of geothermal system. Because the geothermal system combine with the lithology will give information about the water composition.

Response: Thank you for your comment. The hot springs in the study area all belong to the low-medium temperature convective hydrothermal system. The authors have added relevant descriptions and references.

Comment on Result and Discussion:

  1. Line number 113-115, the authors divide the geothermal system into two types coastal and inland geothermal zone. However, the reader cannot distinguish easily in the diagrams. The authors should use different symbol for both type of geothermal type/zone.

Response: Thank you for your comment. The authors have drawn the coastal and inland boundaries in Figure 1. As suggested by the reviewer, the authors found that the analysis of the chemical properties of the hot water in the study area would be more precise if they were divided into coastal versus inland zones. Therefore, the analysis of hot water throughout the text is done by coastal vs. inland instead of by hot springs and geothermal wells, and the relevant figures have been modified accordingly.

  1. The authors described that the samples included the geothermal well water, however there is no indication of the location of the well in the Figure 1. Please provide it. Moreover, the depth of geothermal well is also important to be discussed in addition to the sampling procedure of the geothermal well water.

Response: Thank you for your comment. Four of the 12 hot water samples were taken from geothermal wells, among which the hot water of Yuandanghu(YDH) was found by late drilling (well depth 202.03m), and the remaining three were exposed to natural hot springs in the early days. Then after the hot springs disappeared, people constructed geothermal wells in situ, generally less than 200m deep. The authors have distinguished between hot springs and geothermal wells in Figure 1 and have presented information about geothermal wells, including well depths, in the text.

In addition, the analysis of hot water according to the classification of hot springs and geothermal wells does not show significant differences, and the results are more precise when divided by coastal zone and inland zone. Therefore, the analysis of hot water in the article has been changed to classification by coastal versus inland.

  1. In the geothermal water analysis, commonly SiO2 is described as H4SiO4. The H4SiO4 is the result of the hydrolysis reaction of the silicate minerals. However, in this study the author presented the SiO2 as HSiO3-. Please give the reason.

Response: Thank you for your comment. The laboratory test results were expressed as the concentration of HSiO3-, which we converted to the concentration of SiO2 in the analysis, i.e., dividing the concentration of HSiO3- by 1.2298.

  1. Line 144-146, the author described the wide variation of the stable isotope composition of the hot water that may result from the water-rock interaction or mixing. However, the authors did not specify which one is more dominant. Please also discuss the potential of mixing with sea water.

Response: Thank you for your comment. The authors need to correct the original description that the range of isotopic variation in hot water in the study area is limited. The authors have plotted the seawater mixing line and the water-rock interaction line. The trend of water-rock interaction of hot water is not apparent, and the primary process was mixing hot water and seawater in the coastal zone. In addition, to further illustrate that mixing may have occurred, the authors have placed the cl-isotope correlation analysis, described initially later in this paper, in this section.

  1. The authors should discuss about the method to select the most appropriate geothermometer method.

Response: Thank you for your comment. The authors have added a "Discussion" section to the manuscript, which describes the aspects to be considered in determining the applicability of different methods. In addition, specific methodological recommendations for calculating geothermal reservoir temperatures in different cases are presented.

Comment on Conclusion:

  1. It seems that the did not decide which geothermometer methods is reliable for the studied area.

Response: Thank you for your comment. The authors add a comparative analysis of the calculation results of different methods to the manuscript and propose that the quartz geothermometer (without vapor loss) and the multiple mineral saturation index method are reliable.

Reviewer 2 Report

A case study estimating deep geothermal reservoir temperature in a certain area was presented, and different methods were compared. Overall, this a very relevant study with great significance for geothermal energy. The results presented in this work were found to be interesting and educational. This reviewer has only a few suggestions before publication.

1. Similar studies estimating geothermal reservoir temperature in other areas have already been published. This reviewer recommends mentioning a few of them in the Introduction. Then, the results and findings should be compared to and discussed in the context of earlier work in the literature.

2. The authors should include only the main findings of the study in the Conclusion, and not summarize the entire work.

Reviewer 3 Report

The topic addressed in this article is worthy of investigation and the results showed in the experimental section could be of interest. 

The following changes must be considered before considering this article for publication:

- revise subscripts (SiO2)

- the introduction must be extended, a more detailed state of the art should be provided, together with more related references

- in Section 2, a schematic figure about the study area could be helpful

- Figure 6 must be modified, its resolution is too poor

- the same for Figure 8

Round 2

Reviewer 3 Report

The authors considered all of my suggestions when reviewing their paper; thus, I recommend to accept this article, as it is.

Author Response

Thanks for your valuable comments.